# Synergistic Anti-Tumour Effect of Syk Inhibitor and Olaparib in Squamous Cell Carcinoma: Roles of Syk in EGFR Signalling and PARP1 Activation

**DOI:** 10.3390/cancers12020489

**Published:** 2020-02-19

**Authors:** Duen-Yi Huang, Wei-Yu Chen, Chi-Long Chen, Nan-Lin Wu, Wan-Wan Lin

**Affiliations:** 1Department of Pharmacology, College of Medicine, National Taiwan University, Taipei 100, Taiwan; tyh123kimo@gmail.com; 2Department of Pathology, Wan Fang Hospital, Taipei Medical University, Taipei 116, Taiwan; 1047@tmu.edu.tw; 3Department of Pathology, School of Medicine, College of Medicine, Taipei Medical University, Taipei 106, Taiwan; chencl@tmu.edu.tw; 4Department of Pathology, Taipei Medical University Hospital, Taipei 106, Taiwan; 5Department of Medicine, Mackay Medical College, New Taipei City 251, Taiwan; alvin.4200@mmh.org.tw; 6Department of Dermatology, Mackay Memorial Hospital, Taipei 104, Taiwan; 7Mackay Junior College of Medicine, Nursing, and Management, New Taipei City 252, Taiwan; 8Graduate Institute of Medical Sciences, Taipei Medical University, Taipei 106, Taiwan

**Keywords:** Syk, PARP, EGFR, squamous cell carcinoma

## Abstract

Syk is a non-receptor tyrosine kinase involved in the signalling of immunoreceptors and growth factor receptors. Previously, we reported that Syk mediates epidermal growth factor receptor (EGFR) signalling and plays a negative role in the terminal differentiation of keratinocytes. To understand whether Syk is a potential therapeutic target of cancer cells, we further elucidated the role of Syk in disease progression of squamous cell carcinoma (SCC), which is highly associated with EGFR overactivation, and determined the combined effects of Syk and PARP1 inhibitors on SCC viability. We found that pharmacological inhibition of Syk could attenuate the EGF-induced phosphorylation of EGFR, JNK, p38 MAPK, STAT1, and STAT3 in A431, CAL27 and SAS cells. In addition, EGF could induce a Syk-dependent IL-8 gene and protein expression in SCC. Confocal microscopic data demonstrated the ability of the Syk inhibitor to change the subcellular distribution patterns of EGFR after EGF treatment in A431 and SAS cells. Moreover, according to Kaplan-Meier survival curve analysis, higher Syk expression is correlated with poorer patient survival rate and prognosis. Notably, both Syk and EGFR inhibitors could induce PARP activation, and synergistic cytotoxic actions were observed in SCC cells upon the combined treatment of the PARP1 inhibitor olaparib with Syk or the EGFR inhibitor. Collectively, we reported Syk as an important signalling molecule downstream of EGFR that plays crucial roles in SCC development. Combining Syk and PARP inhibition may represent an alternative therapeutic strategy for treating SCC.

## 1. Introduction

The epidermal growth factor (EGF) receptor/ligand system plays key roles in essential cellular functions, including proliferation and migration. Ligand binding to the epidermal growth factor receptor (EGFR) induces the formation of receptor dimerisation and stimulates multiple pathways of signal transduction. To date, EGFR has been linked to several malignant phenotypes of human cancers, including proliferation, inflammatory response, DNA repair, therapeutic resistance, and poor clinical outcomes in patients with cancer [1,2]. Currently, there are several EGFR antagonists, including humanised neutralising monoclonal antibodies and tyrosine kinase inhibitors clinically used in patients with non-small-cell lung [3] and metastatic colorectal cancers [4]. 

Spleen tyrosine kinase (Syk) is a non-receptor tyrosine kinase that is widely expressed in hematopoietic cells. Syk contains two tandem NH2 terminal Src homology 2 (SH2) domains, multiple tyrosine phosphorylation sites, and a COOH terminal tyrosine kinase domain. The SH2 domains bind phosphorylated immunoreceptor tyrosine-based activation motifs (ITAM), and hence, they couple activated immunoreceptors to multiple downstream signalling pathways [5,6,7]. Anomalous regulation of this kinase could lead to different allergic disorders and antibody-mediated autoimmune diseases. Thus, Syk may serve as a therapeutically relevant target for rheumatoid arthritis, asthma, psoriasis, and allergic rhinitis [8,9,10]. It has been believed for years that Syk function is solely linked to hematopoietic cell signalling. However, more recent studies have indicated a ubiquitous pattern of Syk gene expression. In addition to functioning as the therapeutic target of haematological malignancies [11,12], Syk can function as either a tumour suppressor in breast, colorectal, and gastric cancers or a tumour enhancer in lung, pancreatic, ovarian, and squamous cell carcinoma (SCC) of the head and neck [13]. In cancers of non-immune cells, Syk elicits a pro-survival signal, but can also suppress tumourigenesis by restricting epithelial-mesenchymal transition, enhancing cell-cell interactions, and inhibiting migration [13]. To date, only a few studies have suggested a functional link between Syk and EGFR. In our previous study on human primary normal keratinocytes, we demonstrated that Syk is a downstream signal of EGFR and is involved in negatively regulating keratinocyte differentiation [14].

SCC is an uncontrolled growth of abnormal cells arising in the squamous cells, which compose most of the skin’s upper layers (the epidermis). Similar to other cancer cell types, inhibition of EGFR activity is a potential drug regimen for SCC, which overexpresses EGFR [15,16,17]. A recent in vivo xenograph study suggested a potential therapy involving the use of EGFR monoclonal antibodies in SCC [18]. For better therapeutic efficacy, combination regimens are increasingly under investigation. For example, the EGFR inhibitor, gefitinib (Iressa), can enhance the apoptotic effects of cisplatin in SCC [17]. Clinical trials involving the combination regimens of anti-EGFR agents with immune checkpoint inhibitors are currently in progress for SCC of the head and neck [19,20]. In this study, we explored the role of Syk in EGFR signalling in three SCC cell lines (A431, CAL27, and SAS), analysed Syk expression in oral SCC cancer tissues, and determined the relationship between Syk activity and disease prognosis. Finally, because inhibition of poly(ADP-ribose) polymerase (PARP), a key DNA repair enzyme, could improve the efficacy of radiotherapy in human head and neck cancer [21], we determined the combined effects of the Syk and PARP inhibitors on cell viability in SCC.

## 2. Materials and Methods

### 2.1. Reagents and Antibodies

Dulbecco’s Modified Eagle Medium (DMEM) and trypsin-EDTA were obtained from Gibco (Carlsbad, CA, USA). Foetal bovine serum (FBS) was procured from HyClone (Logan, UT, USA). Penicillin-streptomycin-amphotericin B solution was sourced from Biological Industries (Kibbutz Beit Haemek, Israel). Dulbecco’s Phosphate Buffered Saline (PBS) was obtained from Sigma-Aldrich (St. Louis, MO, USA). Gefitinib was procured from Selleckchem (Houston, TX, USA). R406, PP2, and the p-Syk antibody were purchased from Calbiochem (San Diego, CA, USA). Olaparib was procured from Selleckchem. Recombinant human EGF was obtained from PeproTech (Rocky Hill, NJ, USA). PARP, γH2AX, p-EGFR, p-JNK, JNK, p-p38, p-ERK, p-Lyn, p-Src, p-STAT1, p-STAT3, and p-STAT5 antibodies were obtained from Cell Signaling (Beverly, MA, USA). EGFR, p38, ERK, Lyn, Fyn, Fgr, Src, Syk, STAT1, and STAT3 antibodies were purchased from Santa Cruz (Santa Cruz, CA, USA). Antibody against PAR was obtained from BD Pharmingen (San Diego, CA, USA). The β-actin antibody was procured from NOVUS (Littleton, CO, USA). 

### 2.2. Cell Culture

Human skin SCC (A431), oral SCC (CAL27), and head and neck SCC (SAS) cell lines were cultured in DMEM supplemented with 10% FBS and 1% penicillin-streptomycin-amphotericin B. All cell lines were incubated at 37 °C under a humidified atmosphere of 5% CO_2_ in the air. Prior to stimulation of cells with EGFR ligands or otherwise specified conditions, cells were maintained in the medium without FBS for at least 1 d. 

### 2.3. Immunoblotting

Protein expression was determined in cell lysates by electrophoresis and immunoblotting as previously described [14]. 

### 2.4. Real-tTime PCR

The primer sequence pairs used for quantitative real-time PCR were human β-actin: 5′-CGGGGACCTGACTGACT ACC-3′ and 5′-AGGAAGGCTGGAAGAGTGC-3′; and human IL-8: 5′-ACTGAGAGTGATTGAGAGTGGAC-3′ and 5′-AACCCTCTGCACCCAGTTTTC-3′. The experiment and data analysis were conducted as previously described [22].

### 2.5. Enzyme-Linked Immunosorbent Assay (ELISA)

For the detection of secreted proteins in the cell culture supernatant, a commercial ELISA kit for IL-8 (CXCL8) (R&D Systems, Minneapolis, MN, USA) was used to determine the level of released proteins in accordance with the manufacturer’s protocol.

### 2.6. Confocal Microscopic Analysis

All procedures were conducted at room temperature unless otherwise noted. Cells were fixed with 4% paraformaldehyde in PBS for 20 min. After being rinsed three times with PBS, the fixed cells were permeabilised with 0.2% Triton X-100 in PBS for 20 min. After fixation, the samples were blocked with 4% BSA for 1 h and then incubated with the primary antibody for 2 h at room temperature or overnight at 4 °C after aspiration of the blocking solution. The primary antibody was then removed and specimens were washed three times with PBS. Subsequently, the specimens were incubated with the fluorochrome-conjugated secondary antibody for 1 h in the dark. Following immunostaining, the coverslip was counterstained with 4′,6-diamidino-2-phenylindole (DAPI) and mounted on microscope slides in the dark. Slides were analysed using an LSM 780 confocal microscope (Carl Zeiss Micro Imaging GMbH, Jena, Germany).

### 2.7. Crystal Violet Assay

To evaluate the relative cell numbers, cells were seeded in 96-well plates in medium followed by the indicated treatment. After washing with PBS, cells were fixed with methanol for 10 min at room temperature and stained with 0.1% crystal violet. The relative cell number was determined by measuring the absorbance of the dissolved dye at 540 nm after elution with 33% acetic acid.

### 2.8. Intracellular ATP Assay

Intracellular ATP was determined by a CellTiter-Glo^®^ Luminescent Kit (G7571, Promega Heidelberg, Germany) according to the manufacturer’s protocol. After the indicated drug treatment, the reagent in the kit was added to lyse the cells, and then 150 μL solution was transferred to a 96-well plate for chemiluminescence detection using a luminometer. With background subtraction, the values were normalised to the individual control group being 100%.

### 2.9. MTT Assay

The viability of cells was measured by the turnover of yellow 3-(4,5-dimethyl-2-thiazolyl)-2,5-diphenyl-2*H*-tetrazolium bromide (MTT, Calbiochem) to dark blue formazan by mitochondrial reductase in living cells. A sample of 1 × 10^4^ cells was seeded in 96-well plates for at least 16 h followed by the indicated treatment. At the end of the assay, MTT solution (5 mg/mL in PBS) was added to the medium with a 5-fold dilution for 1 h at 37 °C until a purple precipitate was visible. Cells in the medium were then collected and centrifuged at 800 × rpm for 5 min. After discarding the supernatant, the pellet was dissolved in DMSO. The absorbance (OD_550_–OD_630_) was detected using a spectrophotometer. The absorbance of the control group represented 100% cell viability.

### 2.10. Cell Death Assays by Annexin V/PI Staining and Flow Cytometry

After the indicated treatment, cells were collected and washed with ice-cold PBS. Cells were stained with Annexin V-FITC/PI according to the manufacturer’s instructions (BioLegend, San Diego, CA, USA). After incubation, the cells were measured by flow cytometry (FACSCalibur, BD, Franklin Lakes, NJ, USA).

### 2.11. Tissue Samples and Immunohistochemistry

Pathology files of the Wan Fang Hospital (Taipei, Taiwan) from 2000 to 2010 were searched for SCC of the oral cavity. The pathological diagnoses were microscopically reconfirmed by pathologists (WYC and CLC). Tissue microarrays of oral SCC were manually constructed. Triplicate tissue cores of SCC and one tissue core from the adjacent non-neoplastic squamous epithelium were obtained for each case. In total, 105 patients were included in this study. The clinical records were reviewed for tumour recurrence and survival. For immunohistochemistry, the sections were deparaffinised, rehydrated, and blocked with 3% hydrogen peroxide. Heat-induced antigen retrieval was performed in a citric acid buffer (pH 6.0) at 121 °C for 10 min using a decloaking chamber (Biocare Medical, Concord, CA, USA). The sections were incubated with rabbit polyclonal phospho-Syk (Y525/526) antibody (Catalogue No. AP3271a, 1:300; Abgent Inc., San Diego, CA, USA) at 4 °C overnight. Next, the sections were incubated with a prediluted biotin-conjugated secondary antibody (Starr Trek Universal HRP Detection System; Biocare Medical) at room temperature for 30 min, followed by prediluted streptavidin-horseradish peroxidase complex at room temperature for 10 min. The antigen was detected by adding 3,3′-diaminobenzidine. The sections were then counterstained by haematoxylin. The appropriate positive and negative controls were included in these assays. The intensity of p-Syk expression in carcinoma cells was evaluated by immunostaining. The scoring of expression in tumour tissue was performed according to a four-tiered approach: negative (0), weak (1+), moderate (2+), and strong (3+) staining. Based on these criteria, p-Syk expression was defined as either low (scores 0 and 1+) or high expression (scores 2+ and 3+). Tissue samples were obtained and used according to protocols approved by the Taipei Medical University-Wan Fang Hospital Joint Institutional Review Board (approval no. 99049). The study was conducted according to the Declaration of Helsinki principles.

The clinicopathological characteristics and p-Syk expression were compared using the Chi-square test for categorical data and the two-tailed Student’s t test for continuous data. The overall survival and disease-free curves were calculated using the Kaplan-Meier method, and the difference between the low and high p-Syk expression groups was evaluated using the log-rank test. The Cox proportional-hazard model was utilised to identify statistically significant clinicopathological factors affecting the prognosis of patients. A *p* value < 0.05 was considered statistically significant. All statistical calculations were performed using SPSS statistics 17.0 software (SPSS Inc. Chicago, IL, USA).

### 2.12. Statistical Analysis 

Except immunohistochemistry data as mentioned above, all other data were presented as the mean ± standard error of the mean (S.E.M) from at least three independent experiments. Comparisons between two groups were performed using an unpaired two-tailed Student’s *t* test. 

## 3. Results

### 3.1. SFK-Dependent Syk Activation Mediates Downstream Signal Pathways of EGFR in SCC

After observing the crucial role of Syk in skin keratinocytes [14], we further addressed the roles of Syk in skin tumour development. To this end, we first detected the expression levels of Syk and Src family kinases (SFKs) (c-Src, Lyn, Hck, and Fgr) in A431 SCC, melanoma A375, and SK-MEL-28 cells. We found that Syk expression was highly expressed in A431 cells but not in the others. Lyn and Hck expression was also higher in A431 cells than that in melanoma cells. In contrast, c-Src expression was comparable in these cell lines, whereas Fgr was non-detectable (Figure 1A). To confirm the absence of Fgr in these cells, we used U937 monocytes as a comparison. We found that Syk and Lyn were expressed in U937 cells to the same extent as in the A431 cells; however, U937 cells expressed Fgr, but not c-Src or Hck (Figure 1A).

Next, we found that EGF (100 ng/mL) can increase Syk and c-Src phosphorylation in a time-dependent manner. Moreover, constitutive Lyn phosphorylation was detected but was not altered upon EGF stimulation (Figure 1B). Because of the non-availability of a specific antibody for the phosphorylated Hck, we could not determine its activation status in EGF-stimulated A431 cells. Notably, non-selective SFK inhibitor PP2 (10 μM) decreased the phosphorylation of c-Src, Lyn, Syk, and EGFR (Figure 1B). Because PP2 also reduced EGFR phosphorylation, we suggest there is a feedback loop to control EGFR activation through SFKs as previously reported in fibroblasts [23], breast [23,24], and colon cancer cells [25] in a transactivation manner. All these findings suggested that Syk is a signal molecule for EGFR, and its activation is downstream of SFKs.

Next, we used the Syk inhibitor, R406, to test its effects on EGF-induced signalling pathways. As shown in Figure 1C, R406 treatment significantly inhibited the phosphorylation extent of JNK, p38 MAPK, STAT1, and STAT3 in EGF-stimulated A431 cells. Notably, R406 also reduced EGFR phosphorylation as did PP2 (Figure 1C). 

In addition to A431 cells, we also examined the role of Syk in CAL27 (oral SCC) and SAS (head and neck SCC) cells. As observed in A431 cells, EGF stimulated the phosphorylation of EGFR, c-Src, JNK, and p38 in time- and concentration-dependent manners in both the cell lines (Figure 2A). Moreover, EGF activated Syk in CAL27 (Figure 2B) and SAS cells (Figure 2C). In CAL27 cells, EGF-induced phosphorylation of EGFR, STAT3, Akt, and JNK was inhibited by R406 (Figure 2B). Similarly, EGF-induced STAT3 phosphorylation in SAS cells was reduced by R406 (Figure 2C). These findings indicated that Syk is the upstream signal molecule that transduces the downstream signalling pathways of EGFR.

### 3.2. Syk Mediates EGF-Induced IL-8 Upregulation But Is Not Involved in the Regulation of Cell Fate 

Given that IL-8 is a crucial player in cancer development [26], we examined the effects of R406 on EGF-induced IL-8 expression in A431 cells. We found that EGF can upregulate IL-8 mRNA expression and increase IL-8 protein production (Figure 3A), and these actions were inhibited by R406 and gefitinib. Similarly, in SAS and CAL27 cells, IL-8 mRNA induced by EGF was markedly reduced by R406 and gefitinib (Figure 3B,C).

Using BrdU incorporation as an index of cell proliferation, our data did not show the significant effects of EGF and R406 on BrdU uptake in A431 cells (Figure 4A). We also used MTT, ATP, and crystal violet assays to assess the effects of EGF and R406 on cell viability. Our data revealed non-significant effects of both agents on the cell viability of A431 cells (Figure 4B). When Syk mRNA was silenced by the siRNA approach, EGF still could not affect cell viability in A431 cells (Figure 4C). We also tested another EGFR activator, TGFα and found that in agreement with the data for EGF, TGFα (10 and 100 ng/mL), in the absence or presence of R406, did not change cell viability as assessed by MTT (Figure 4D) and crystal violet assays (Figure 4E) in A431 cells. Similarly, MTT assays revealed that R406 did not alter the viability of SAS and CAL27 cells (Figure 4F).

### 3.3. Syk Regulates Intracellular EGFR Movement in a Cell Type-Specific Manner 

In our previous study, in NHEK cells, we observed the co-localisation of EGFR and Syk. Moreover, Syk was moved along with EGFR trafficking intracellularly after EGF stimulation [14]. Here, we further detected the intracellular localisation of EGFR in A431 and SAS cells before and after EGF stimulation. Unlike NHEK cells, where EGFR during the resting state was apparently present in the membrane, EGFR in A431 cells was present in both the cytosol and plasma membrane during the resting state. After EGF stimulation, EGFR was translocated to the plasma membrane and peri-nuclei of a spot like. The latter site was co-stained with the early endosomal marker EEA1, suggesting the occurrence of common EGFR trafficking from the internalisation to the late endosome/lysosomal degradation and endocytotic recycling pathways. The treatment with the Syk inhibitor R406 reduced the membrane location but not the peri-nuclear translocation of EGFR induced by EGF (Figure 5A). In SAS cells, EGFR was primarily observed in the plasma membrane as NHEK and could move to the peri-nuclei early endosome upon EGF stimulation. R406 treatment appeared to induce an uneven distribution of EGFR in peri-nuclear sites after EGF stimulation in SAS cells (Figure 5B).

### 3.4. Syk Activity Is Associated with Clinicopathologic Features and Outcome of Patients with Oral SCC

To better understand the prognostic role of Syk in oral SCC, an immunohistochemical analysis for p-Syk expression was performed on the tissue microarray sections of oral SCC, and the clinicopathological phenomena and p-Syk expression in tumour tissues in 105 patients were analysed. We first compared the expression levels of p-Syk in oral SCC with that of matched non-neoplastic squamous epithelium (Figure 6A). Oral SCC had stronger p-Syk expression than normal squamous epithelium. The association between clinicopathological characteristics and p-Syk expression is shown in Table 1. Oral SCC in the high p-Syk expression group tended to be poorly differentiated in grading (*p* = 0.002). Lymphovascular invasion was more commonly detected in oral SCC with high p-Syk expression (*p* = 0.018). The associations between p-Syk and other clinicopathological factors, such as gender, age, perineural invasion, tumour size and extent, lymph node metastasis, and pathologic staging were not statistically significant. In addition, the Kaplan-Meier curve analysis revealed that the overall survival of the patients with low p-Syk expression was significantly longer than that of the patients with high p-Syk expression (*p* = 0.022) (Figure 6B). Patients with low p-Syk expression tended to have long disease-free survival; however, this was not statistically significant (*p* = 0.13) (Figure 6C). Univariate and multivariate analyses using the Cox proportional-hazard model were conducted to determine prognostic factors affecting overall survival (Table 2 and Table 3). In the univariate analysis, old age (>65 years), larger tumour and/or extent (T3 + T4), presence of lymph node metastasis, high pathological staging (stage III and IV), and high p-Syk expression were associated with a poorer prognosis. The multivariate analysis showed that old age (>65 years), larger tumour and/or extent, and high p-Syk expression were independent poor prognostic factors for overall survival.

### 3.5. Syk and EGFR Regulate PARP1 Activation and Syk Inhibitor Exerts a Synergistic Anti-Tumour Effect with Olaparib

Because we found that Syk is involved in EGFR signalling in SCC, we intended to determine if Syk can serve as a therapeutic target. Moreover, PARP1 is involved in DNA damage repair and its expression is negatively correlated with patient survival rate and prognosis. Given that PARP1 inhibitors (PARPi) are currently used clinically for treating ovarian and metastatic breast cancers [27,28], and PARP inhibition results in the radiosensitisation of HPV/p16-positive HNSCC cells [21,29], we also aimed to determine the combined effects of Syk and PARP inhibition. First, we found that both R406 and the EGFR inhibitor, gefitinib, could induce PARP activation and DNA damage in the three SCC cell lines as indexed by increased PAR formation and γH2AX expression, respectively. Furthermore, PARP cleavage was induced by both agents (Figure 7A). Secondly, we found that although R406 and olaparib did not significantly affect cell viability, their combination induced significant cell death (Figure 7B). Such an enhanced cytotoxic l effect was more apparent in SAS and CAL27 cells than that in A431 cells. Moreover, because targeting EGFR has been evaluated in SCC [15,16,17], we tested the combined effects of gefitinib and olaparib. We found that although gefitinib itself did not affect cell survival, as demonstrated in Figure 4F, co-treatment with both the agents increased cell death, whereas the efficacy was slightly weaker than that of R406/olaparib (Figure 7B). Altogether, these results suggested a promising treatment of SCC by the combined inhibition of Syk and PARP.

## 4. Discussion

The importance of EGFR in tumourigenesis has been well established. Excessive activation of EGFR signalling by overexpression of, or mutations in, EGFR has been found in various types of human tumours, making EGFR a widely recognised target for cancer therapy [30]. Indeed, prior studies have established enhanced anti-tumour efficacy of the EGFR inhibitor (or antibody) with radiation [31] or cytotoxic drugs for A431 cells [17,32]. Nevertheless, long-term EGF treatment also leads to drug resistance, which is an urgent issue that must be overcome. This indicates the necessity to explore new targets for anti-tumour therapy. In this study, we found that Syk is a new downstream signal molecule of SFKs in activated EGFR-transduced signalling pathways, and similar to SFKs, Syk possesses a positive feedback loop to enhance EGFR phosphorylation. Moreover, the association of higher Syk activity with poor clinicopathological features of patients with oral SCC, including poor overall survival and high lymphovascular invasion, was observed. Furthermore, a combined inhibition of Syk and PARP leading to enhanced SCC cell death may be developed into a new therapeutic regimen in the future. 

The major functions of Syk in hematopoietic cells involving innate and adaptive immune systems are well characterised. Apart from the established molecular mechanisms and functional roles in autoimmunity, accumulating evidence suggests that Syk plays dual roles in cancers, i.e., either as a tumour suppressor or promoter [13]. Reportedly, the reduced expression of Syk in breast cancer and melanoma cells is associated with a higher degree of malignancy and poor prognosis. The tumour suppressor action of Syk has been associated with positive regulation of p53 activity [33] and p21 expression [34], as well as negative effects on mitotic progression [35,36]. In contrast, Syk serves as a tumour promoter in hematopoietic malignancies and could be a potential oncogenic driver in small cell lung cancer [37] and ovarian cancer [36] by inducing pro-survival signals [13]. Therefore, Syk could become a new therapeutic target in some cancer cell types. For example, a recent study indicated that a Syk inhibitor could sensitise TRAIL-induced apoptosis by downregulating Mcl-1 in breast cancer cells [38]. In this study, we strengthened our previous speculations that Syk is an upstream signalling molecule for EGFR in SCC cell lines and mediates EGF-induced IL-8 gene expression. IL-8 is a potent chemoattractant molecule that performs different pro-tumoural functions, such as proliferation, angiogenesis, and metastasis of cancer cells. Since in patients with cancer, IL-8 is mainly produced by tumour cells, its serum concentration has been shown to be correlated with the tumour burden, and IL-8 is an effective pharmacodynamic biomarker to detect an early response to immunotherapy [26]. The serum level of IL-8 is significantly increased in patients with recurrent and metastatic head and neck SCC [39]. In addition, elevated levels of IL-8 are found in the saliva of patients with oral SCC; thus, IL-8 could serve as a salivary biomarker of oral SCC [40]. In addition, using human oral SCC samples, we demonstrated the strong association of Syk activity with poor disease progression. All these data prompt us to suggest Syk inhibition as an alternative cancer treatment. 

In addition to clarifying the role of Syk in EGFR signalling and the prognosis of patients with SCC, we tested the combined effects of Syk and PARP inhibition on cancer cell death. The PARP1 inhibitor has been shown to sensitise radiotherapy in head and neck SCC [21]. Moreover, because cetuximab can augment cytotoxicity with PARP inhibition in head and neck SCC [41], we also tested the anti-tumour activity of gefitinib and olaparib. Although PARP1 inhibition has been applied in clinical settings in combination with different therapeutic agents, it is still crucial to develop alternative regimens to overcome drug resistance and insufficient efficacy in cancer therapy. In addition, some recent studies have indicated the effect of EGFR activation on the regulation of PARP1 in a cell context-dependent manner. In human LNCaP prostate cancer cells, radiation-induced PARP activation is enhanced through EGFR-ERK signalling [42]. In UT-SCC5 and SAS head and neck cancer cells, PARP1 has been shown to serve as a mediator of EGFR/MEK-dependent regulation of DNA double-strand breaks [43]. In breast cancer cells, a contextual synthetic lethality may exist between combined EGFR and PARP inhibitors [44]. In glioblastoma cells, EGFRvIII overexpression causes increased ROS-dependent DNA strand break accumulation and PARP activation, and reduced DNA repair gene expression, including PARP1, results in improved patient survival [45]. In hepatocellular carcinoma, EGFR and c-MET cooperate to enhance resistance to PARP inhibitors [46]. Furthermore, a heterodimer of EGFR and MET can phosphorylate Y907 of PARP1 in the nucleus of hepatocellular carcinoma and contribute to this resistance [46]. In pancreatic cancer cells, adaptive expression of EGF following exposure to ionising radiation may induce radio-sensitisation of cells through the induction of the cyclin D1/p53/PARP pathway [47]. In contrast, EGF in combination with radiation augments the radiation effects in A431 cells by inhibiting DNA damage repair [48]. Additionally, EGFR-mutant lung cancer cells display higher sensitivity to the PARP inhibitor olaparib [49]. All these studies suggest that EGF can regulate PARP1, which might impact the efficacy of cancer therapy, particularly when DNA damage is caused by cancer therapy. However, to date, studies on the relationship between Syk and DNA breaks are limited. The only finding reported is that DNA damage and DNA double-strand breaks can suppress the expression of Syk in B [50] and NK cells [51].

Our data revealed a time-dependent effect of Syk and EGFR inhibition on DNA damage and PARP1 activation in three SCC cell lines. R406 exerts higher responses in terms of DNA damage and PARP1 activation than those exerted by gefitinib in CAL27 and A431 cells. However, such parallel events are not observed in SAS cells where R406 induces greater DNA damage but less PARylation compared to that of gefitinib. Currently, the reasons underlying this paradoxical finding are unknown. Notably, even though PARP1 cleavage was observed upon treatment of R406 or gefitinib, the cell viability assay did not reveal significant effects for either agent on cell death. We speculate that this might be due to the insufficient activation of executive caspase 3 for cell death and/or the existence of other pro-survival pathways. Another possible reason is that the PARP1 cleavage is mediated by other proteases. Previous studies showed that PARP1 can be cleaved by caspase 1 and caspase 7 to yield 85–89 kD PARP-1 fragments [52,53,54], and such cleavage leads to an NF-κB-dependent inflammatory gene expression [54]. Therefore, PARP1 cleavage is a hallmark of apoptosis yet not essential. In addition, we speculate the PARP1 activation evoked by R406 and gefitinib alone might participate in the cell survival, for example as previously observed for the DNA repairing process upon moderate DNA damage. However, interestingly, enhanced cell death effect was observed upon the co-inhibition of Syk and PARP. Notably, this enhanced cell death effect is greater than that induced by gefitinib and olaparib. Therefore, we suggest that Syk inhibition leads to the interruption of EGFR-dependent and -independent cellular actions that might exert synergistic effects with olaparib in terms of cell death. It is necessary to further address the molecular mechanisms underlying the coordinative cellular events mediated by Syk and PARP1 in the future. Additionally, it will be interesting to investigate and compare the efficacies between the Syk inhibitor and EGFR mAb (e.g., cetuximab) in SCC upon co-treatment with PARP1 inhibitors.

A431 cells compose a high EGFR-activating skin SCC. Notably, it is unexpected to find EGF at concentrations of 50–100 ng/mL, which is mitogenic for general cell types and cannot induce cell proliferation and viability in A431 cells within 24 h of incubation. Supporting this notion, previous studies reported a growth arrest effect of EGF in A431 cells that was observed slowly [55,56]. To date, the mechanism through which EGF induces growth arrest in A431 cells has been ascribed to the upregulation of IRF-1 through SFR-dependent STAT1 and STAT3 activation [57,58]. Similarly, cell lines that hyperexpress EGFR, such as MDA-MB-468 cells, have been documented to undergo receptor-mediated apoptosis through STAT3 [59]. Therefore, because of the overexpression of EGFR and constitutive overactivation of EGFR in A431 cells, we speculate there might be an aberrant regulation mechanism in the EGFR-dependent cellular functions. Another distinct feature in A431 is the subcellular distribution of EGFR. Unlike primary keratinocytes and SAS cells, EGFR is not only localised in the plasma membrane but is also present in the cytosol in A431 cells. Moreover, after EGF stimulation, the plasma membrane level of EGFR is increased in A431 cells, which is opposite to the general trafficking direction of activated EGFR. We found lower amounts of EGFR expression in SAS cells compared to that of A431 cells could more rapidly undergo EGFR internalisation after EGF stimulation for 1 h. All these phenomena observed in A431 cells might result from the high abundance of EGFR expression in this specific cell line. Another cell type-specific action of the Syk inhibitor is its effects on EGF-induced EGFR trafficking. We found that in A431 cells, R406 could decrease the plasma membrane level of EGFR upon EGF stimulation, whereas in SAS cells, EGF-induced EGFR trafficking to the early endosome displayed a higher level upon Syk inhibition. Because Syk can associate with EGFR in keratinocytes [14] and altered EGFR internalisation and recycling can regulate drug sensitivity in cancer cells [60,61], we plan to further determine the role of Syk in EGFR trafficking, including receptor internalisation, receptor recycling to the plasma membrane, and degradation by lysosomes in the future.

## 5. Conclusions

We confirmed our previous findings in keratinocytes that Syk mediates EGFR signalling in SCC and Syk is a positive contributor to disease progression in patients with oral SCC. Even though Syk and PARP inhibition cannot induce cell death individually, their combination causes enhanced cell death. Thus, our data provide the rational for co-treatment of PARP inhibition and Syk inhibition in SCC.

## Figures and Tables

**Figure 1 cancers-12-00489-f001:**
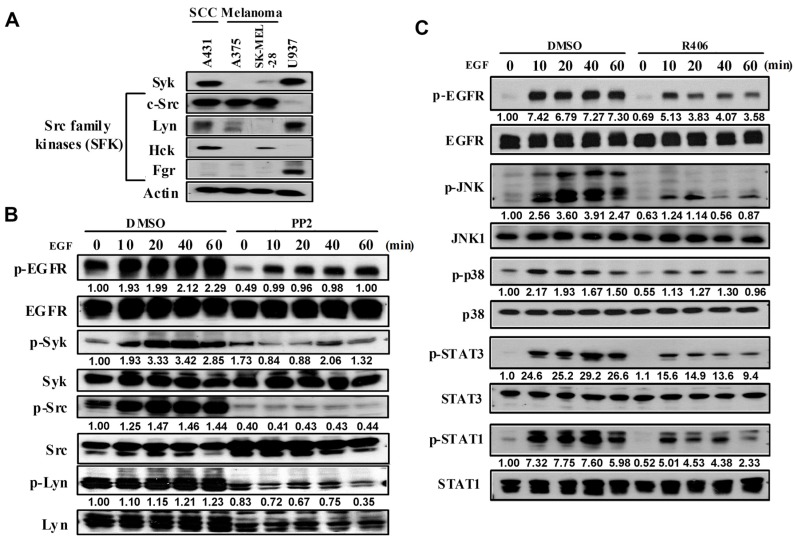
EGF can induce Syk activation through SFK signalling in A431 cells. (**A**) Western blot analysis of Syk and SFK expression in various cancer cell lines. (**B**) Phosphorylation of EGFR, Syk, Src, and Lyn after EGF (100 ng/mL) treatment with or without PP2 (10 μM) for 0, 10, 20, 40, and 60 min in A431 cells. (**C**) Phosphorylation of EGFR downstream signalling molecules after EGF (100 ng/mL) treatment with or without R406 (1 μM) for 0, 10, 20, 40, and 60 min in A431 cells. Data are presented from three independent experiments.

**Figure 2 cancers-12-00489-f002:**
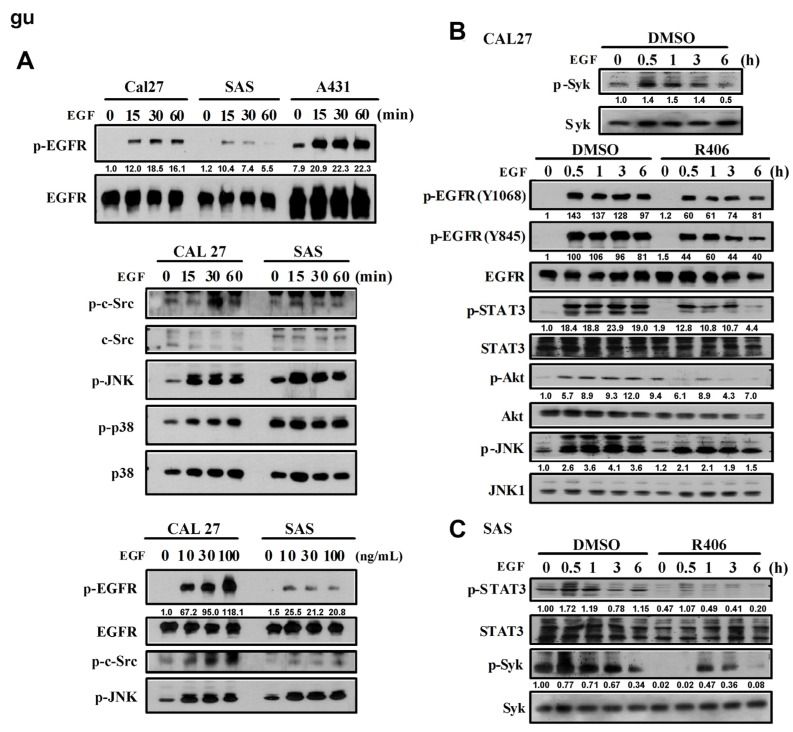
Syk is also an upstream signalling molecule of EGFR in SCC. (**A**) EGF-induced phosphorylation of EGFR and downstream signalling molecules for 0, 15, 30, and 60 min after EGF (100 ng/mL) treatment compared to that of A431. Concentration-dependent EGF treatment (0, 10, 30, and 100 ng/mL) for 30 min in CAL27 and SAS cells. (**B**,**C**) Phosphorylation of EGFR and downstream signalling molecules after EGF (100 ng/mL) treatment with or without R406 (1 μM) for 0, 0.5, 1, 3, and 6 h in CAL27 (**B**) and SAS cells (**C**). Data are presented from three independent experiments.

**Figure 3 cancers-12-00489-f003:**
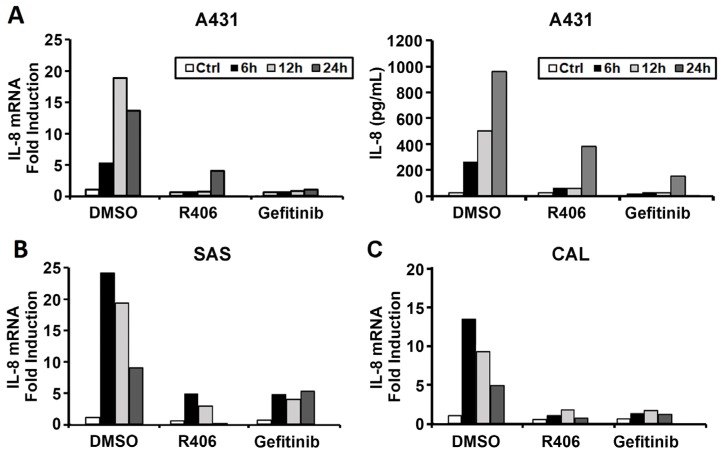
Syk mediates EGF-induced IL-8 mRNA and protein expression in SCC. EGF (100 ng/mL) treatment in A431 (**A**), SAS (**B**), and CAL27 (**C**) cells for 0, 6, 12, and 24 h and the addition of R406 (1 μM) or gefitinib (5 μM) as a positive control. IL-8 mRNA levels were determined by RT-PCR and protein levels by ELISA.

**Figure 4 cancers-12-00489-f004:**
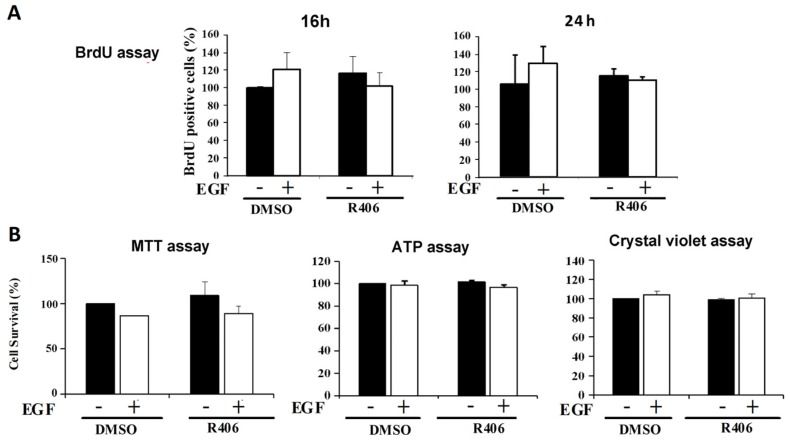
Syk is not involved in the proliferation and viability of SCC cells. (**A**) Effects of R406 (1 μM) with or without co-treatment with EGF (50 ng/mL) on A431 cell proliferation for 16 and 24 h were analysed by BrdU assay. (**B**) Effects of R406 (1 μM) with or without co-treatment with EGF (100 ng/mL) on A431 cell viability for 24 h analysed by MTT, ATP, and crystal violet assays. (**C**) Effects of Syk siRNA with or without EGF (50 ng/mL) for 24 h on A431 cell viability analysed by an MTT assay. (**D**,**E**) Cell viability of A431 cells after treatment with R406 (1 μM) with or without co-treatment with TGFα (10 or 100 ng/mL) for 24 h analysed by MTT and crystal violet assays. (**F**) Viability of SAS and CAL27 cells treated with R406 for 24 h determined by MTT assay. Data are expressed as the mean ± S.E.M. from three independent experiments.

**Figure 5 cancers-12-00489-f005:**
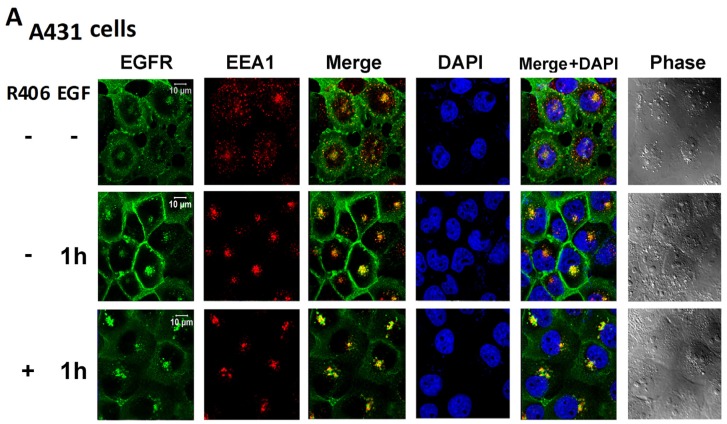
Syk modulates EGFR localisation under EGF treatment in A431 and SAS cells. Confocal microscopic analyses of EGFR and EEA1 (early endosome marker) localisation after treatment of EGF (50 ng/mL) for 1 h with or without the addition of R406 (1 μM) in A431 (**A**) and SAS cells (**B**). Data are from three independent experiments.

**Figure 6 cancers-12-00489-f006:**
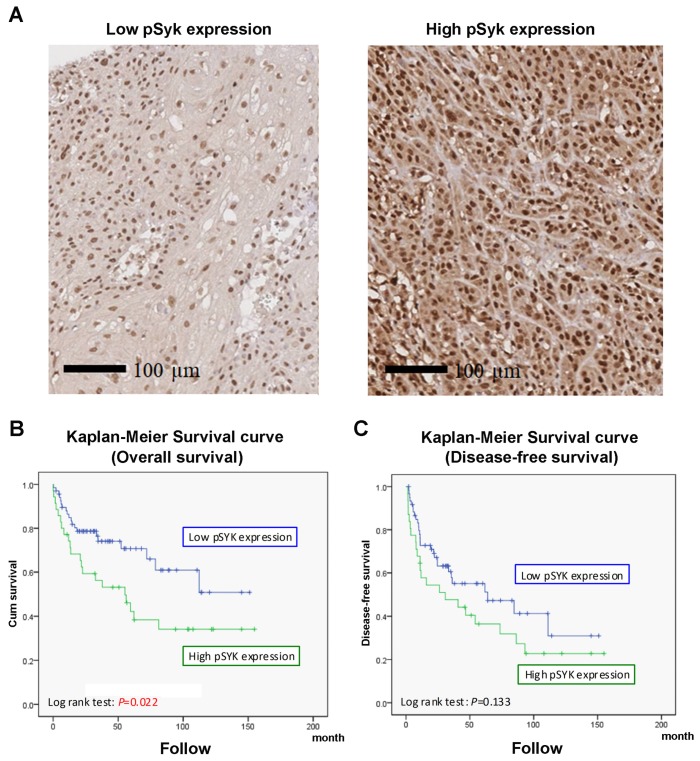
Syk is important for SCC development in human patients. (**A**) Immunohistochemical analysis of p-Syk expression in tissue microarray sections of oral SCC. (**B**,**C**) Correlation of Syk activity to overall survival (**B**) and disease-free survival (**C**) in patients with oral cancer.

**Figure 7 cancers-12-00489-f007:**
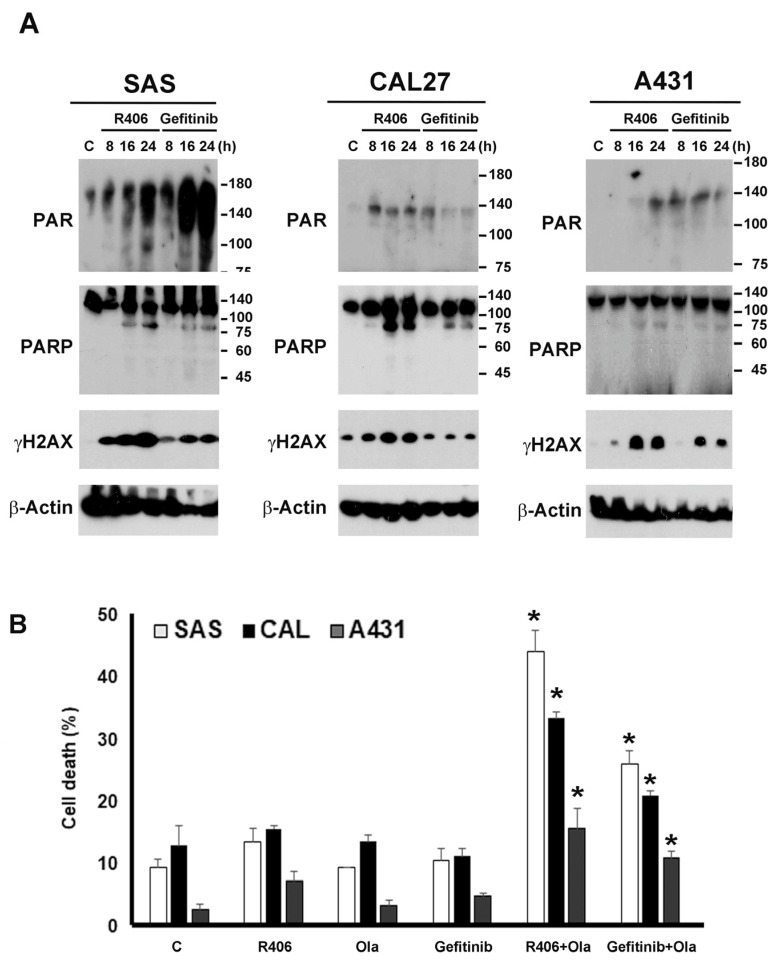
R406 and gefitinib induce PARP activation, and inhibition of Syk and PARP enhances cell death in SCC. (**A**) Cells were treated with R406 (1 μM) or gefitinib (5 μM) for different periods, Figure 1. γ-H2AX, and β-actin. (**B**) Cells were treated with R406 (1 μM), gefitinib (5 μM) and/or olaparib (Ola, 10 μM) for 24 h. Cell death was determined by Annexin V/PI staining and flow cytometry. Data in (**B**) are expressed as the mean ± S.E.M. from three independent experiments. * *p* < 0.05 indicates a synergistic effect of R406 + Ola and gefitinib + Ola on cytotoxicity compared to individual drug treatments.

**Table 1 cancers-12-00489-t001:** Association between clinicopathological characteristics and p-Syk expression.

Characteristics	P-Syk Expression		*p* Value
	Low (*n* = 68)	High (*n* = 37)	
Gender			0.488
Male	60 (88%)	35 (95%)	
Female	8 (12%)	2 (5%)	
Mean Age (years) ± SD	53.5 ± 14.5	56.7 ± 12.1	0.255
Age			0.628
<65 years	51 (75%)	30 (81%)	
≧65 years	17 (25%)	7 (19%)	
Grading of SCC			0.002
Well to moderately differentiated	58 (85%)	21 (57%)	
Poorly differentiated	10 (15%)	16 (43%)	
Lymphovascular invasion			0.018
Negative	49 (72%)	17 (47%)	
Positive	19 (28%)	19 (53%)	
Perineural invasion			0.153
Negative	35 (51%)	13 (36%)	
Positive	33 (49%)	23 (64%)	
Tumor size and extent			1.000
T1+T2	46 (68%)	25 (68%)	
T3+T4	22 (32%)	12 (32%)	
Lymph node metastasis			0.669
Negative	45 (67%)	23 (62%)	
Positive	22 (33%)	14 (38%)	
Stage (AJCC 7th Ed)			1.000
I + II	33 (50%)	18 (49%)	
III + IV	33 (50%)	19 (51%)	

Abbreviation: AJCC, American Joint Committee on Cancer.

**Table 2 cancers-12-00489-t002:** Univariate analysis of clinicopathological factors and p-Syk expression affecting overall survival.

Variables	Overall Survival		*p* Value
	Hazard Ratio	95% CI	
Gender			0.388
Male	1	-	-
Female	1.509	0.593–3.840	
Age			0.013
<65 years	1	-	-
≧65 years	2.257	1.191–4.277	
Grading of SCC			0.537
Well-moderately differentiated	1	-	-
Poorly differentiated	1.216	0.653–2.266	
Lymphovascular invasion			0.608
Negative	1		
Positive	1.173	0.637–2.158	
Perineural invasion			0.423
Negative	1		
Positive	1.288	0.694–2.393	
Tumor size and extent			<0.001
T1+T2	1	-	-
T3+T4	4.636	2.562–8.389	
Lymph node metastasis			0.002
Negative	1	-	-
Positive	2.606	1.432–4.742	
Stage (AJCC 7th Ed)			<0.001
I + II	1	-	-
III + IV	4.053	2.104–7.808	
P-Syk expression			0.025
Low	1	-	-
High	2.024	1.094–3.742	

Abbreviations: CI, confidence interval; AJCC, American Joint Committee on Cancer.

**Table 3 cancers-12-00489-t003:** Multivariate analysis of clinicopathological factors and p-Syk expression affecting overall survival.

Variables	Overall Survival		*p* Value
	Hazard Ratio	95% CI	
Gender			0.279
Male	1	-	-
Female	1.887	0.597–5.960	
Age			0.001
<65 years	1	-	-
≧65 years	4.153	1.774–9.724	
Grading of SCC			0.393
Well-moderately differentiated	1	-	-
Poorly differentiated	0.710	0.324–1.558	
Lymphovascular invasion			0.809
Negative	1		
Positive	1.108	0.480–2.557	
Perineural invasion			0.586
Negative	1		
Positive	1.246	0.566–2.743	
Tumor size and extent			0.015
T1+T2	1	-	-
T3+T4	3.429	1.267–9.276	
Lymph node metastasis			0.070
Negative	1	-	-
Positive	2.572	0.927–7.139	
Stage (AJCC 7th Ed)			0.668
I + II	1	-	-
III + IV	1.356	0.337–5.447	
P-Syk expression			0.002
Low	1	-	-
High	3.393	1.581–7.283	

Abbreviation: CI, confidence interval; AJCC, American Joint Committee on Cancer.

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
