# Peer review of "Synergistic Anti-Tumour Effect of Syk Inhibitor and Olaparib in Squamous Cell Carcinoma: Roles of Syk in EGFR Signalling and PARP1 Activation"

_cancers, 2020, doi:10.3390/cancers12020489_

Round 1

Reviewer 1 Report

This manuscript presents the synergetic effects of Syk inhibitor and PARP inhibitor in SCC cells in vitro, and also showed the relationship between p-Syk expression and overall survival. 

I like this manuscript very much because it shows quite interesting results both in vitro and clinical samples. I strongly believe that every basic research should feedback to the clinical area, and this manuscript shows fascinating results for both researchers and clinicians. 

Moreover, Syk and Syk have just come for the spotlight and this manuscript has a good point to make. 

I believe this manuscript would be suitable for publication in this version. Of course, authors should modify it in accordance with other reviewers. 

Author Response

N/A

Reviewer 2 Report

Authors evaluated Syk phosphorylation as a prognostic factor in oral SCC and analyzed the interest of targeting the non-receptor tyrosin kinase in different SCC cell lines in vitro, alone or in combination with a PARP inhibitor.

While p-Syk expression as a prognostic factor is convincing, I have major concerns with the rationale of the study and the in vitro experiments.

A- From a (pre-)clinical point of view :

1/ In HNSCC, the clinical use of mAbs targeting EGFR yields a better overall response than TKI targeting the same receptor/pathway, alone or in combination. Did the authors expect a better response to Syk inhibition in SCC or a specific subgroup of patients ?    

2/ Combination of EGFR targeted therapies, especially mAbs such as Cetuximab, and PARP inhibitors has already been evaluated in HNSCC and other cancers. The authors should have tested this combination and explain how Syk inhibition could improve tumor cell response to PARP inhibition.

3/ At least two in vivo models (one "low p-Syk" and one "high p-Syk") should have been tested, either on ectopically xenografted cell lines or patient-derived xenografts.  

B- From a technical point of view :

1/ How many independent repeats were performed for the in vitro experiments  ? This should be mentioned for all methods and error bars should be added to Figure 3.

2/ Differents pharmacological inhibitors were tested in vitro (PP2, R406, Gefitinib, Olaparib) at the same concentrations for the 3 SCC cell lines. Authors should explain how the drug concentrations were chosen.  

3/ For Western Blot figures : signal saturation is observed for different bands/proteins (Figure 1B : p-EGFR, EGFR, Syk, p-Src, Src, p-Lyn, Lyn ; Figure 1C : EGFR, p-JNK, p-STAT1, STAT1; Figure 2A : EGFR; Figure 2B : STAT3, p-JNK; Figure 2C : STAT3, p-Syk), please include densitometry readings.

4/ Syk inhibition by R406 had no significant effect on A431 cell proliferation and viability.  Did the authors observe the same results with SAS and CAL27 cell lines ?

5/ Compared to R406 treatment alone, the effect of combination with PARP inhibitor Olaparib was evaluated by a different technical approach (Annexin V staining). Did the authors confirm apoptosis induction by a proliferation and/or survival assay ?   

6/ IL8 expression was analyzed at the mRNA +/- protein levels in SCC cell lines. Why did the authors focus on IL8 ? Does IL8 inhibition by Gefitinib or R406 impact tumor cell migration in vitro and/or angiogenesis in vivo ? This should have been evaluated or at least discussed in the paper.

7/ EGFR trafficking upon EGF stimulation +/- Syk inhibition was studied in A431 and SAS cells. What is the purpose of comparing EGFR trafficking in both cell lines ? How does it impact tumor cell sensitivity to Syk and/or EGFR inhibition ? This should be explained.   

8/ Syk phosphorylation in tumor tissue samples was evaluated by immunohistochemistry. The authors should explain how the cut-off between low and high expression was set. 

9/ Figure 6C : "Overall survival" should be replaced by "Disease-free survival" 

Author Response

A- From a (pre-)clinical point of view :

1/ In HNSCC, the clinical use of mAbs targeting EGFR yields a better overall response than TKI targeting the same receptor/pathway, alone or in combination. Did the authors expect a better response to Syk inhibition in SCC or a specific subgroup of patients?    

Ans: I agree with the reviewer’s comment that EGFR TKIs are less effective, less specificity and higher toxicity compared to EGFR mAbs (Gougis et al., 2019). Because Syk is a tumor promoter in SCC and our data suggest the association of Syk activity to poor progression of SCC, we think Syk inhibitor might be considered for the treatment of SCC patients. While Syk inhibitor has been clinically used in B cell lymphoma and allergic rhinitis, its potential use in cancer is just at a beginning stage. More basic studies supporting clinical benefits and clinical trials are still needed. In addition, compared to the cytotoxic effect in combination with PARP1 inhibitor, Syk inhibition has higher efficacy than iressa. Therefore, our study points out a potential anti-cancer strategy and more studies are still needed to prove this possibility.

2/ Combination of EGFR targeted therapies, especially mAbs such as Cetuximab, and PARP inhibitors has already been evaluated in HNSCC and other cancers. The authors should have tested this combination and explain how Syk inhibition could improve tumor cell response to PARP inhibition.

Ans: Thanks for the suggestion. Instead using mAbs of EGFR we have tested the combination of iressa and olaparib. We found that iressa and olaparib also exhibit a synergistic cytotoxicity in 3 SCC cell lines, but the anti-tumor efficacy is less compared to R406 plus olaparib (Fig. 7B). We have added these new data and discussion in our revised manuscript.

3/ At least two in vivo models (one "low p-Syk" and one "high p-Syk") should have been tested, either on ectopically xenografted cell lines or patient-derived xenografts.  

Ans: This is a constructive suggestion by conducting the xenograft experiment in mice to strengthen the tumor promoting role of Syk in SCC. However, due to the time limitation for manuscript revision, we feel sorry not being able to conduct this experiment. We believe our data obtained from SCC cells as well as cancer patients should support our conclusion.

B- From a technical point of view:

1/ How many independent repeats were performed for the in vitro experiments? This should be mentioned for all methods and error bars should be added to Figure 3.

Ans: Thanks for the comment. We have added statistical analysis in the Methods and figure legends.

2/ Different pharmacological inhibitors were tested in vitro (PP2, R406, Gefitinib, Olaparib) at the same concentrations for the 3 SCC cell lines. Authors should explain how the drug concentrations were chosen.  

Ans: Based on previous reports including ours, we used 10 mM PP2 (J Invest Dermatol, 2016, 136, 192; Mol Cell Biochem, 2011, 348, 11; J Biol Chem, 2005, 280, 6036; Blood, 2013, 122, 4911), 1 mM R407 (J Leukocyte Biol, 2015, 97, 825; J Invest Dermatol, 2016, 136, 192;  Cancers 2019, 11, 202; Pharm Res, 2014, 31, 3060; Cell Death Disease, 2019, 10, 358), 5 mM Gefitinib (J Invest Dermatol, 2016, 136, 192; Cancer Medicine, 2019, Dec 11, 001–10) and 10 mM Olaparib (Cell Reports 2019, 27, 3422; Cancers, 2019, 11, 1373; BMC Cancer, 2016, 16,725; Scientific Reports, 2017, 7, 12876). All these concentrations are within their respectively proper concentration ranges for studies in cells. 

3/ For Western Blot figures: signal saturation is observed for different bands/proteins (Figure 1B: p-EGFR, EGFR, Syk, p-Src, Src, p-Lyn, Lyn ; Figure 1C : EGFR, p-JNK, p-STAT1, STAT1; Figure 2A : EGFR; Figure 2B : STAT3, p-JNK; Figure 2C : STAT3, p-Syk), please include densitometry readings.

Ans: We have included the quantification data for each phosphorylated proteins as suggested in Fig. 1B, 1C, 2A, 2B and 2C.  

4/ Syk inhibition by R406 had no significant effect on A431 cell proliferation and viability. Did the authors observe the same results with SAS and CAL27 cell lines?

Ans: We found no effects of R406 on cell viability in SAS and CAL27 cells based on MTT or Annexin V/PI staining, and these new findings were shown in Fig. 4F and Fig. 7B.

5/ Compared to R406 treatment alone, the effect of combination with PARP inhibitor Olaparib was evaluated by a different technical approach (Annexin V staining). Did the authors confirm apoptosis induction by a proliferation and/or survival assay?   

Ans: In the cell viability experiment of drug combination, we used Annexin V/PI staining and the data were shown in Fig. 7B. Although MTT assay is a simple and easy way to present cell viability based on the succinate dehydrogenase activity, it might not be completely related to cell viability due to the altered mitochondrial activity under some stress conditions. We believe Annexin V/PI staining is more convincing for cell survival assay than others. 

6/ IL8 expression was analyzed at the mRNA +/- protein levels in SCC cell lines. Why did the authors focus on IL8? Does IL8 inhibition by Gefitinib or R406 impact tumor cell migration in vitro and/or angiogenesis in vivo? This should have been evaluated or at least discussed in the paper.

Ans: Thanks for the comment. We have discussed the pro-tumor functions of IL-8, in particular in SCC patients. Because IL-8 is mainly produced by tumor cells and its levels in serum and saliva can be biomarkers of SCC, we used it as a downstream effect of EGFR to verify the benefits of Syk inhibitor in cancer therapy.

7/ EGFR trafficking upon EGF stimulation +/- Syk inhibition was studied in A431 and SAS cells. What is the purpose of comparing EGFR trafficking in both cell lines? How does it impact tumor cell sensitivity to Syk and/or EGFR inhibition? This should be explained.   

Ans: Our previous study indicate that Syk can interact with EGFR in keratinocytes in plasma membrane and traffic with EGFR to early endosome upon EGF stimulation. Because altered EGFR internalization and recycling can regulate drug sensitivity in cancer cells, here we examined whether Syk can regulate EGFR trafficking in SCC. We found that after EGF stimulation, Syk inhibition can increase EGFR internalization in A431 cells and increase EGFR localization in early endosome in SAS cells (Fig. 5). We suggest Syk might be able to regulate EGFR trafficking. The discussion on these findings is supplemented.

8/ Syk phosphorylation in tumor tissue samples was evaluated by immunohistochemistry. The authors should explain how the cut-off between low and high expression was set. 

Ans: Thanks for the comment. We have added the information to define the low and high expression of Syk phosphorylation in tumor tissues from SCC patients in the “Material and Method” section.

9/ Figure 6C: "Overall survival" should be replaced by "Disease-free survival" 

Ans: Thanks for the correction. We have revised the title of Fig. 6C.

Reviewer 3 Report

The manuscript by Huang et al. describes the role of Syk during SCC specific EGFR signaling. Specifically, they show that EGF can phosphorylate Src family members (SFK) and that inhibition of SFKs or Syk blunts EGFR activation. The Syk inhibition does not block proliferation or survival of SCC cell lines but their EGF mediated IL8 production and EGFR cellular localization. Furthermore high pSyk expression in oral SCC sections correlates with poor patient survival. Inhibition of Syk or EGFR induces DNA damage and PARP activation in vitro and combination of Syk inhibition with PARP inhibition enhanced SCC cell death, which indicates a possible therapeutic anchor point.

Their study builds up on a previous publication on EGFR and Syk during keratinocyte differentiation. Also the interplay of EGFR and SFKs has been already studied extensively, which takes away some of its novelty.

Several important controls are missing.

Major points:

To what extend does SCC pSyk expression compare to that of normal human skin?

Figure 1C lacks plots for pSyk, pSrc and pLyn as controls for inhibitor specificity.

Figure 4 lacks Iressa as control for EGFR induced cell proliferation and/or survival in these cells.

Figure 7B lacks Iressa and Iressa and Olaparib as control.

The manuscript could be improved by adding an in vivo mouse model of SCC (or xenograft models from the used cancer cell lines) treated with R406 and Olaparib in Figure 7.

Minor points:

English should be checked throughout the manuscript

Author Response

Major points:

To what extend does SCC pSyk expression compare to that of normal human skin?

Ans: From our study Syk-p levels at resting state in SCC cell lines and keratinocytes do not show significant difference, but EGF stimulation all can induce significant Syk-p.  

Figure 1C lacks plots for pSyk, pSrc and pLyn as controls for inhibitor specificity.

Ans: From the data shown in Fig. 1B, we suggest Syk is a downstream signal molecule of SFKs. We also observed similar effect in EGF-stimulated keratinocytes that PP2 can inhibit EGF-induced Src phosphorylation (Wu et al., 2016). In ITAM-containing immunoreceptors, Syk is the major upstream kinase, and its activation is controlled by SFKs (Lowell, 2011, Src-family and Syk kinases in activating and inhibitory pathways in innate immune cells: signaling cross talk. Cold Spring Harb Perspect Biol. 2011;3(3). pii: a002352). Based on these well recognized information that Syk is downstream molecule of SFKs, we do not further determine the effect of Syk inhibitor on SFKs.

Figure 4 lacks Iressa as control for EGFR induced cell proliferation and/or survival in these cells.

Ans: We have added new data of iressa in Fig. 7B. We found that Iressa treatment alone does not significantly affect cell viability in these SCC cell lines.

Figure 7B lacks Iressa and Iressa and Olaparib as control.

Ans: Thanks for the comment. We have included data of iressa ± olaparib in Fig. 7B. We found iressa and olaparib also display the synergistic anti-tumor effect in three SCC cell lines. However, compared to R406 with olaparib, the cell death extent is less.

The manuscript could be improved by adding an in vivo mouse model of SCC (or xenograft models from the used cancer cell lines) treated with R406 and Olaparib in Figure 7.

Ans: This is a constructive suggestion to strengthen the antitumor effect of R406 and Olaparib in mice with SCC. However, due to the time limitation for manuscript revision, we feel very sorry not being able to conduct this experiment. However, we hope the data from SCC cells to human patients can provide strong evidence to reveal the role of Syk in SCC cancer progression.

Minor points:

English should be checked throughout the manuscript

Ans: We have revised our manuscript by a native English speaker.

Round 2

Reviewer 2 Report

Minor points :

1/ Authors have to explain how the drug concentrations were chosen specifically for A431, SAS and CAL27 cell lines. What does "proper concentration ranges for studies in cells" mean ? For example, cells were treated with 5µM Gefitinib (Fig 7B), and the authors give 2 different references :

J Invest Dermatol 2016 : NHEKs are treated with Gefitinib (10µM)  Cancer Medicine 2019 :  cannot find the reference (volume 8, issue 17 or 18 ?)

Does it mean that NHEK, A431, SAS and CAL27 have the same IC50 for EGFR inhibition ? 

2/ For Western-Blot figures, densitometry readings and relative quantification cannot be evaluated with signal saturation

3/ Authors suggest that MTT assay "might not be completely related to cell viability", but A431 cells viability was evaluated by an MTT assay in Fig.4. Why did the authors change the technical approach between Fig 4 and Fig 7B ? 

4/ Iressa should be replaced by Gefitinib

Major point :

As previously mentioned, results presented in Fig7B should be supported by in vivo experiments, comparing a low- and a high phospho-Syk model (CTRL; R406 alone; Olaparib alone; R406+Olaparib).

Author Response

Minor points:

1/ Authors have to explain how the drug concentrations were chosen specifically for A431, SAS and CAL27 cell lines. What does "proper concentration ranges for studies in cells" mean? For example, cells were treated with 5 µM Gefitinib (Fig 7B), and the authors give 2 different references:

J Invest Dermatol 2016: NHEKs are treated with Gefitinib (10 µM)  Cancer Medicine 2019 : cannot find the reference (volume 8, issue 17 or 18 ?)

Does it mean that NHEK, A431, SAS and CAL27 have the same IC50 for EGFR inhibition? 

Ans: (1) Attached in the right hand site is the information of reference (Cancer Medicine 2019 Dec 11. doi: 10.1002/cam4.2772.) that we mentioned before. (2) Because EGFR responses in these cell lines are all sensitive to gefitinib, the concentration of gefitinib (5 µM) we used is based on that commonly used.  We do not mean that NHEK, A431, SAS and CAL27 have the same IC50 for EGFR inhibition. Here we show 3 more references in using this concentration of gefitinib.

ANTICANCER RESEARCH, 2009, 29, 5023-5032. (5-10 uM)

Cancer Biology & Therapy, 2011, 11:11, 927-937.(10 uM)

Journal of Experimental & Clinical Cancer Research, 2019, 38, 254. (5-10 uM)

2/ For Western-Blot figures, densitometry readings and relative quantification cannot be evaluated with signal saturation

Ans: We understand the limitation of immunoblotting regarding the saturation on immune intensity. We are sure that our signals provided do not have such condition.

3/ Authors suggest that MTT assay "might not be completely related to cell viability", but A431 cells viability was evaluated by an MTT assay in Fig.4. Why did the authors change the technical approach between Fig 4 and Fig 7B? 

Ans: We used MTT, ATP and crystal violet assays (Fig. 4) to show the same finding of R406 on viability as measured by Annexin V/PI staining (Fig. 7B).

4/ Iressa should be replaced by Gefitinib

Ans: As suggestion, we have changed “iressa” to “gefitinib”.

Major point:

As previously mentioned, results presented in Fig.7B should be supported by in vivo experiments, comparing a low- and a high phospho-Syk model (CTRL; R406 alone; Olaparib alone; R406+Olaparib).

Ans: Thanks the comment of Editors.

Academic Editor Notes Dear Authors,

Please amend the manuscript according to the suggestions of the Reviewers. For the next round of revision you do not need to perform the animal experiments, as human data is already presented in the manuscript.

Reviewer 3 Report

The authors have revised the manuscript and added the necessary additional controls in Fig.7B as requested.
However, the authors refrained from adding the requested in vivo experiment of the anti-tumor effect of R406 and Olaparib in mice due to time limitations for manuscript revision.

Major point:

This is a very important experiment and this reviewer cannot accept the manuscript without the in vivo data. 

Minor points:

Iressa is misspelled in Figure 7B.

Mean is misspelled (mane) in 2.12. Statistical analysis.

Details about the statistical tests (e.g. anova or students t test) used throughout the manuscript are necessary for each Figure and should also be indicated in the materials and methods section (2.12. Statistical analysis). P values for the indicated stars (e.g. *P<0.05) should be added to each figure and the comparison group should be also indicated in the figures (e.g. a line connecting the compared bars under the star).

Author Response

Major point:

This is a very important experiment and this reviewer cannot accept the manuscript without the in vivo data. 

Ans: Thanks the comment of Editors.

Academic Editor Notes Dear Authors,

Please amend the manuscript according to the suggestions of the Reviewers. For the next round of revision you do not need to perform the animal experiments, as human data is already presented in the manuscript.

Minor points:

Iressa is misspelled in Figure 7B.

Ans: Thanks for the correction, and we have changed it.

Mean is misspelled (mane) in 2.12. Statistical analysis.

Ans: Thanks for the correction, and we have changed it.

Details about the statistical tests (e.g. anova or students t test) used throughout the manuscript are necessary for each Figure and should also be indicated in the materials and methods section (2.12. Statistical analysis). P values for the indicated stars (e.g. *P<0.05) should be added to each figure and the comparison group should be also indicated in the figures (e.g. a line connecting the compared bars under the star).

Ans: Thanks for the suggestion, and we have changed it. The statistical significance symbol was explained in the figure legend of Fig. 7B. Because the symbol indicates the synergistic effects of R409+Ola and Gefitinib+Ola as compared to the groups of R406, Ola and Gefitinib individually, it would be too complicated to add many lines for comparison.